# Data-Based Variables Used as Indicators of Dairy Cow Welfare at Farm Level: A Review

**DOI:** 10.3390/ani11123458

**Published:** 2021-12-04

**Authors:** Barbara Lutz, Sibylle Zwygart, Christina Rufener, Joan-Bryce Burla, Beat Thomann, Dimitri Stucki

**Affiliations:** 1Centre for Proper Housing of Ruminants and Pigs, Federal Food Safety and Veterinary Office FSVO, Agroscope Tänikon, 8356 Ettenhausen, Switzerland; Christina.Rufener@agroscope.admin.ch; 2Clinic for Ruminants, Vetsuisse Faculty, University of Bern, Bremgartenstrasse 109a, 3012 Bern, Switzerland; sibylle.zwygart@vetsuisse.unibe.ch (S.Z.); dimitri.stucki@vetsuisse.unibe.ch (D.S.); 3Farm-Animal Welfare Division, Agroscope Tänikon, 8356 Ettenhausen, Switzerland; joan-bryce.burla@agroscope.admin.ch; 4Veterinary Public Health Institute, Vetsuisse Faculty, University of Bern, 3012 Bern, Switzerland; beat.thomann@vetsuisse.unibe.ch

**Keywords:** cattle, herd, wellbeing, welfare, health, indicator, routine herd data, records, assessment

## Abstract

**Simple Summary:**

In recent years, the interest in the use of data from routine herd records in the monitoring of dairy welfare at farm level has increased. This review compiles 13 papers to outline the current potential of data-based variables for animal welfare monitoring. All the identified studies showed associations between data-based variables and farm-level dairy cow welfare and therefore provide a first indication of the possible use and suitability of data-based variables for welfare monitoring. However, we found that the definitions of animal welfare, its assessment on the farm, and the data-based variables varied considerably. Consequently, the current state of research does not allow a conclusive assessment of the potential of data-based variables for animal welfare monitoring. Therefore, future research is needed to clarify the potential of data-based variables. Harmonisation of the data-based variables and the use of valid measurements that reflect the multidimensionality of welfare could contribute to increased comparability between the studies.

**Abstract:**

During the last years, the interest in data-based variables (DBVs) as easy-to-obtain, cost-effective animal welfare indicators has continued to grow. This interest has led to publications focusing on the relationship between DBVs and animal welfare. This review compiles 13 papers identified through a systematic literature search to provide an overview of the current state of research on the relationship between DBVs and dairy cow welfare at farm level. The selected papers were examined regarding their definition of animal welfare and classified according to this definition into three categories: (a) papers evaluating DBVs as predictors of animal welfare violations, (b) papers investigating the relationship between DBVs and animal-based measurements, and (c) papers investigating the relationship of DBVs to scores of welfare assessments like the Welfare Quality protocol or to overall welfare scores at farm level. In addition, associations between DBVs and indicators of animal welfare were extracted, grouped by the type of DBV, and examined for replications that may confirm the associations. All the identified studies demonstrated associations between DBVs and animal welfare. Overall, the first indications of a possible suitability of DBVs for predicting herds with animal welfare violations as well as good or poor animal welfare status were given. The evaluation of relationships between DBVs and animal-based measurements (ABMs) found mortality-based DBVs to be frequently associated with ABMs. However, owing to varying definitions of animal welfare, the use of different variants of DBVs, and different methods used to assess DBVs, the studies could only be compared to a limited extent. Future research would benefit from a harmonisation of DBVs and the use of valid measurements that reflect the multidimensionality of welfare. Data sources rarely investigated so far may have the potential to provide additional DBVs that can contribute to the monitoring of cow welfare at farm level.

## 1. Introduction

Over the past decades, the increasing concerns of society and consumers regarding the wellbeing of farm animals has led to the development of a variety of animal welfare programmes and certification systems. In addition to governmental initiatives, these programmes and certification systems include private quality assurance systems, such as animal welfare labels and organic schemes initiated by producer organisations, retailers, and non-governmental organisations [1,2,3]. Consequently, the need to reliably assess the animal welfare status at farm level—e.g., by means of external audits—is becoming more and more important.

However, animal welfare is a complex multidimensional concept for which no universal definition exists. The current understanding of animal welfare goes beyond the biological functioning and also includes behaviour and the emotional status of the animals [4]. For example, in the concept of the “five freedoms” [5] good welfare requires freedom from hunger and thirst, freedom from discomfort, freedom from pain, injury or disease, freedom to express normal and natural behaviours, and freedom from fear and stress.

Due to its complexity, animal welfare itself cannot be measured directly but must be reflected through a variety of measurements that represent the multidimensionality [6]. The measurements and indicators used to assess on-farm animal welfare are typically categorised as resource- or management-based parameters and animal-based parameters [7,8]. Resource-based parameters describe the environment that affects the animals—e.g., the type of housing or the supply of resources [8]—and are thus related only indirectly to animal welfare [3]. In contrast, animal-based measurements (ABMs)—e.g., body condition score or disease state—directly reflect the health and welfare status of the animals [8] as a result of husbandry and management practices [9]. ABMs are therefore widely accepted in welfare science [10,11,12] and have been integrated into many animal welfare assessment methods [13,14,15,16], including some for dairy cattle [14,17]. A disadvantage of on-farm ABMs is their time-consuming survey [8,9,18], requiring alternative indicators if larger numbers of farms are to be assessed. A promising approach is to complement or replace on-farm ABMs with data-based variables (DBVs). In this context, DBVs that are based on data or records collected directly from the animals are also referred to as indirect animal-based measurements [11]. For the DBVs used to monitor animal welfare, data routinely collected on the animals—e.g., records from identification and registration, data on milk yield and quality, and animal health parameters—are suitable [9,19]. For dairy cows, an abundance of data are available in European countries as harmonised EU legislation provides for records on birth, movements, and death for each individual bovine animal [20]. In addition to these identification and registration records, other data routinely collected from cows—such as data on milk yield and quality or animal health parameters—are also suitable as DBVs [9,19]. However, to be suitable for animal welfare assessment, the DBVs must be easy to record, and a clear relationship between the DBV and the welfare status of the animals must be demonstrated [21].

With regard to dairy herd welfare, the ongoing interest in DBVs has led to numerous scientific publications investigating the suitability of DBVs as animal welfare indicators. With our review, we compile this work to provide an overview of the current state of research by following two objectives. First, we describe the definitions of animal welfare used in 13 papers identified through a literature search and categorise the publications according to these definitions. Second, we review the selected papers and extract the associations of DBVs and the indicators of animal welfare at farm level to outline the potential of DBVs for monitoring animal welfare.

## 2. Design and Results of the Literature Search

The basis for this review was a systematic literature search for a project aimed at identifying and establishing DBVs for dairy welfare in Switzerland. For this purpose, in August 2019 and December 2020, five scientific literature databases were screened (PubMed, Web of Science, Scopus, CAB Direct, and ScienceDirect). For the searches, synonyms for ‘dairy cows’ were combined with different animal welfare and data-associated terms (Table 1).

The queries were mainly carried out as title-and-abstract searches; in the case of large numbers of hits (>100), only titles and keywords were searched instead. Furthermore, filters were used to exclude papers related to human health. Identified literature published since 1995 in German, English, and French was subjected to a screening by the first author. In this process, it was checked whether the literature met the criteria to be included in the present review: first, the use of variables based on routine herd data and second, the investigation of the relationships between these variables and dairy welfare at herd or farm level. From the screening, 13 papers met the criteria and were selected as a basis for the present review (Table 2).

To provide an overview of the papers, both the data sources used to construct DBVs and the definitions and methods used to determine the animal welfare status at farm level were described (Section 3). Subsequently, the papers were categorised according to the definition of welfare used, and the current state of research was reviewed for each welfare definition category (see Section 4, Section 5 and Section 6). If the methodology of the papers was comparable, the associations between DBVs and indicators of animal welfare were extracted from the papers and grouped by type of DBV. Consideration was given to whether the associations were based on univariable or multivariable analyses. The strengths of the associations were not included in this review because the studies differed in terms of their conditions and analyses. In this review, the names of the specific DBVs and their variants are written in italics to distinguish them from data sources and umbrella terms that may include several related DBVs.

## 3. Overview of the Origin of Data-Based Variables and the Animal Welfare Definitions Used

The 13 selected publications used different data sources to extract and calculate suitable potential DBVs (Table 3). Overall, the databases consisting of identification and registration records were most frequently used as a source of DBVs, whereas disease and treatment data as well as records on herd disease status and reasons for culling were rarely included in the analyses. In total, more than 150 variables were found from such databases and examined as possible indicators of animal welfare.

Although all 13 selected papers examined links between DBVs and animal welfare and thus pursued the same objective, they differed in the definition and assessment of the animal welfare status at farm level. For background information on the assessment of animal welfare at farm level, see Box 1.

Three of the papers defined poor welfare as violations of legal animal welfare requirements (see Section 4). Five studies examined the relationships between DBVs and ABMs (see Section 5). Finally, seven publications, including two also included in Section 5, investigated the relationship between DBVs and animal welfare scores composed of several ABMs, e.g., Welfare Quality (WQ) criteria and principles, the area scores of the Centro di Referenza Nazionale per il Benessere Animale (CReNBA) welfare protocol, and overall scores (see Section 6).

Box 1How can animal welfare be assessed at farm level?The complexity of animal welfare and the lack of a universal definition make it difficult to assess the animal welfare status, especially at farm level. Over the past decades, various assessment methods have been developed, e.g., the ‘animal needs index’ in Austria [35], the ‘Centro di Referenza Nazionale per il Benessere Animale’ (CReNBA) protocol in Italy [36], the ‘Kuratorium für Technik und Bauwesen in der Landwirtschaft e. V.’ (KTBL) animal welfare protocol in Germany [37], and the Welfare Quality (WQ) protocol as a European approach [17]. At the present time, none of these assessment protocols can serve as a ‘gold standard’ for surveying animal welfare, but the WQ protocol is the most extensive approach [38].The welfare assessments differ mainly in the number and type of measurements used. For example, the ‘animal needs index’ assesses available resources, whereas the WQ protocol focuses on animal-based measurements (ABMs) and is only complemented by resource- and management-based measurements when ABMs are not available. Further differences occur in whether and how the results of specific measurements and indicators are combined (and often weighted) to quantify the animal welfare status at farm level. In the KTBL animal welfare protocol, for example, the different indicators are evaluated separately. In contrast, other assessment protocols such as the CReNBA protocol and the WQ protocol aim to combine the results of the measurements into an overall score.For the CReNBA animal welfare assessment, measurements from four areas are carried out: farm management and personnel, facilities and equipment, animal-related measures, and microclimatic environmental conditions and alarm systems. To determine the numeric overall score, the results of the different measurements are weighted according to their relevance and summed up. Consequently, the farm-level welfare status identified by the CReNBA protocol can be presented using area scores as well as an overall score.The WQ protocol provides a hierarchical integration process for which, in a first step, the specific measurements (mostly ABMs) are grouped into 12 criteria according to their interrelationships. Within the criteria, the measurements are weighted according to relevance and can partially compensate each other. The 12 criteria themselves are designed to cover all relevant aspects of the WQ animal welfare definition and are combined—again weighted—into four principle scores: ‘good feeding’, ‘good housing’, ‘good health’, and ‘appropriate behaviour’. From these principle scores, an overall score can be calculated that classifies the welfare status at farm level as ‘excellent’, ‘improved’, ‘acceptable’, and ‘not classified’. Thus, with the WQ protocol, the welfare at farm level can be presented using data of the raw measurements, increasingly aggregated levels of criteria and principles, and an overall score.

## 4. Data-Based Variables as Predictors of Farms Violating Animal Welfare

In this section, the focus will be on the three papers that stand out by defining poor animal welfare as the presence of violations against the animal welfare legislation. Among these papers, two Irish studies investigated whether there are similarities in DBVs in herds with officially confirmed animal welfare violations [22,23], whereas a Danish study analysed correlations between DBVs and two common animal welfare violations [24].

Kelly et al. [22,23] focused on the identification and validation of DBVs as indicators of animal welfare violations. In both studies, 18 cattle herds, including five dairy herds with officially confirmed welfare conflicts, were used as case herds. In this context, animal welfare incidents were defined as situations in which the responsible person inflicted avoidable pain or suffering on the animals or did not act appropriately to prevent it, as well as situations where there was no rapid response to animals in pain or suffering. For the first study [22], six DBVs that had been selected during previous work and by expert opinion were calculated on an annual basis over a 4- to 9-year period that included the animal welfare violation. For the analyses, it was investigated whether the case herds showed any parallels in terms of performance in the DBVs. The four variables *late registration of calves*, *on-farm burial of carcasses*, *increasing number of movements to knackeries,* and *movements to unknown herds* were prominent in the case farms. In contrast, no patterns emerged for the DBVs *changes in herd size* and *number of calves registered per cow and year*. The follow-up study [23] aimed to validate the four variables identified in the previous study and the additional variable *movements to factories or abattoirs* by comparing the distribution of the DBVs between the case herds with confirmed welfare violations and the remaining Irish herds. For all five variables, this study revealed a significant difference in distribution between these two groups. Furthermore, the DBVs were tested alone and in combination at different cut-offs for their ability to distinguish between herds with and without violations of animal welfare. Because of the low sensitivities and specificities, the authors did not consider any of the DBVs or sets to be applicable to identify farms with violations of animal welfare.

A further approach by Otten et al. [24] pursued the same objective, but they limited animal welfare violations to the two most commonly found in Danish animal welfare inspections: the presence of sick or injured animals not kept in sick pens and the presence of animals in a condition requiring euthanasia. Out of 73 farms visited, 23 were classified as case herds because they met at least one of the two violations. For this study, 25 variables were examined for their association with animal welfare violations. In a univariable analysis, associations were found for five variables at a statistical significance level of α = 0.2: *yield for first lactation cows*, *standard deviation (SD) of milk yield for first lactation cows, SD of milk yield for second lactation cows*, *number of veterinary treatments per 100 cow years,* and the *number of abattoir remarks*. By backward elimination, a final multivariable model for predicting herds with violations of animal welfare was obtained consisting of three variables: *increasing SD of milk yield for first lactation cows*, *high bulk milk somatic cell count (≥250,000 cells per millilitre)*, and a suspiciously *small number of veterinary treatments (≤25 treatments per 100 cow years)*.

In summary, the Irish and Danish approaches differ fundamentally in the DBVs used, the methodology, and the definition of animal welfare violations. Nevertheless, all three publications emphasise that DBVs have the potential to contribute to animal welfare monitoring in the future. The three studies presented provide a first evaluation of DBVs as predictors of animal welfare violations, which needs to be complemented by future research. As no evaluation of specific DBVs is possible at this stage, we suggest for future research the inclusion of a broad set of clearly defined DBVs. Additionally, to correctly interpret the results, the specific animal welfare violations investigated should be outlined. Whereas the application of DBVs as predictors of welfare violations requires a validation in the specific reference population, a broad set of studies in different countries could provide important information on underlying correlations between DBVs and animal welfare of dairy farms.

## 5. Data-Based Variables and Animal-Based Measurements

This section is based on five publications in which DBVs were evaluated on individual measurements or sets of individual measurements—predominantly ABMs and only a few resource- or management-based measurements. In contrast to the work presented in Section 4, DBVs from similar data sources as well as comparable ABMs were used, which allowed us to extract and compare the associations identified in these studies.

### 5.1. Review of Studies

In 2009, a Swedish study [25] aimed to identify herds with poor animal welfare by using a set of DBVs. The on-farm welfare status was defined by five ABMs in dairy: lameness, injuries or inflammation, rising behaviour, cleanliness, and body condition, with the last two measures also collected for calves and young stock. Univariable analysis between 65 DBVs and the ABMs revealed associations for 28 DBVs, of which 18 were also confirmed in a multivariable analysis. Most of all, DBVs reflecting fertility status and mortality of different age groups were frequently associated with ABMs.

In two studies, de Vries et al. [27,28] investigated the suitability of DBVs for predicting the farm-level welfare status. For the first study [27], the results of 22 on-farm WQ measurements on 196 farms were dichotomised to identify whether a severe problem was present at farm level or not. Relationships between the measurements and 46 DBVs were identified using logistic regression. Additionally predictive models of DBVs were built to predict whether a farm had a severe problem in the specific measurement. These models achieved a performance, expressed as area under the curve (AUC), between 0.57 for the model predicting the ‘avoidance distance index’ and 0.94 for the model for ‘access to at least two drinkers’. DBVs based on mortality in different lactation groups, especially *mortality within the first 60 days of lactation*, were included most frequently in the predictive models. In contrast, the second study [28] included only six ABMs. An initial univariable analysis between the DBVs and ABMs revealed a large number of associations at a significance level of α = 0.2. In addition, predictive models for the performance in the ABMs were built using DBVs and housing data. Again, the predictive model for the ‘avoidance distance index’ achieved only a poor accuracy. In contrast, the models for the WQ measurements of severely lame cows, cows with lesions or swellings, cows with dirty hindquarters, very lean cows, and the frequency of displacements all achieved a moderate accuracy.

Two more studies that explored associations between DBVs and ABMs were done by Otten et al. [29,31]. For the paper published in 2016 [31], an index consisting of DBVs was validated against an animal-based approach including twelve ABMs. DBVs were calculated for the periods covering 365, 180, and 90 days preceding the on-farm assessment and analysed for their association with the ABMs. Most of the associations were found for DBVs based on milk yield data, slaughterhouse data, and mortality data. The associations between the ABMs and these DBVs could not be demonstrated consistently for all three calculation periods.

The more recent study by Otten et al. [29] focused exclusively on lameness prevalence, i.e., a single ABM. For this study, lameness assessments were carried out on 40 dairy herds, of which herds with a lameness prevalence of ≥16% (third quartile) were classified as having a high lameness prevalence. The four DBVs *cow mortality*, *bulk milk somatic cell count*, *lean cows at slaughter*, and *SD of first calving age* were calculated as annual means and analysed for their association with lameness prevalence. Significant associations were found for both continuous and dichotomised *cow mortality* and *bulk milk somatic cell count*. Conversely, *lean cows at slaughter* showed associations with lameness prevalence only as dichotomised DBV. The DBV *cow mortality* allowed the identification of herds with high prevalence of lameness with an AUC of 0.76.

### 5.2. Detailed Description of the Associations between Data-Based Variables and Animal-Based Measurements

For a DBV to be suitable as an indicator of animal welfare, it must not only be easily accessible, but the link to animal welfare must be clearly demonstrated. If associations between DBVs and ABMs are shown in several studies, this congruence may confirm the association and thus may indicate the value of DBVs for animal welfare monitoring. Therefore, in this section we focus on associations that have repeatedly been demonstrated.

#### 5.2.1. Mortality

Variables based on cow mortality were used in many studies (Table 4). As already noted by Thomsen et al. [21], it is often not clear whether emergency-slaughtered cows are included in the mortality calculation. Furthermore, some of the studies used the overall cow mortality, whereas other studies used mortality in different lactation periods. In the following, the associations that emerged for the different variants of cow mortality with specific ABMs were grouped together.

It was particularly notable that several DBVs related to mortality were associated with the proportion of lame cows. An increased *mortality rate (%)* for the three different animal groups *cattle aged >2 years* [28], *cows 0–60 days in milk (DIM)* [27], and *cows 120–210 DIM* [27] manifested itself in an increased proportion of severely lame cows. Additionally, the *on-farm mortality of cattle >210 DIM (%)* was increased in herds that showed a higher prevalence of moderately lame cows [27]. Furthermore, the DBV *cow mortality* alone could identify herds with high prevalence of lameness [29].

Thus, whereas a clear pattern emerged for the associations between mortality and lameness, the remaining associations provided less clear findings. *On-farm mortality of cattle >2 years (%)* showed a univariable [28] and a multivariable positive association with the percentage of cows with lesions or swellings [27]. Furthermore, the *on-farm mortality of cattle* in both periods *0–60 DIM (%)* and *120–210 DIM (%)* was positively associated with the frequency of collisions while lying down [27]. Associations with DBVs related to mortality were also repeatedly shown for other ABMs, but these were often oppositely directed and came only from the studies by de Vries et al. [27,28].

#### 5.2.2. Calf Mortality

Three studies revealed correlations between the DBVs related to calf mortality and the proportion of lean cows in the herd (Table 5). The variables *calf mortality (90 days)* [31] and *calf mortality 2–6 months* [25] were both positively associated with the proportion of very lean cows. In addition, a univariate association was found between the variable *on-farm mortality of cattle aged 0–3 days (%)* and this ABM [28].

A connection between the DBVs related to calf mortality and cow lameness was also demonstrated in several studies. An increased rate of *stillbirths* was associated with an increased prevalence of moderately lame cows [27], and *calf mortality (90 days)* was increased in herds with a higher proportion of severely lame cows [31]. Furthermore, in a univariable analysis, *on-farm mortality of cattle aged 0–3 days (%)* was shown to be associated with the prevalence of severely lame cows [28].

For the specific variables studied, as with cow mortality, the age groups considered varied widely. One study did not specify which age group was defined as a calf and thus taken into account for calf mortality [31]. The variable *stillbirth* also lacked the exact definition of which calves were considered. Usually, a stillbirth is defined in veterinary medicine and agriculture as a calf that is stillborn after a normal gestation period, dies at birth or dies within 12–48 h after birth. However, in the studies identified [25,27], it was not stated which period after birth was considered, or whether a calculation deviating from the common definition was chosen.

#### 5.2.3. Herd Size

From the databases containing identification and registration records, the total number of dairy cows kept on a given farm—i.e., the total herd size—can be calculated. All associations that resulted for DBVs related to herd size are summarised in Table 6. For *herd size* itself, univariable and multivariable positive relationships were shown with each of the ABMs for cows with dirty hindquarters, the proportion of very lean cows [28], and the prevalence of severely lame cows [27,28]. The variable *changes in herd size* was also associated with the proportion of severely lame cows in a univariable analysis. Furthermore, a multivariable analysis revealed that a prevalence of severely lame cows >11.9% was negatively associated with *change in herd size* [28].

The two papers presented here examined the relationships between herd size and ABMs without considering causality or confounding factors. Beyond the work presented here, the question of the relationship between herd size and welfare status is highly contested in animal welfare research. Robins et al. [39], who studied this question intensively for their review, concluded that there was little evidence for a simple relationship between animal welfare and herd size. With this in view, the associations we were able to identify for herd size, which come from only two studies based on the same data set, should be interpreted with caution.

#### 5.2.4. Individual-Cow Somatic Cell Count

The somatic cell counts (SCCs) collected on individual animals are of great importance for monitoring udder health status, both at animal and at herd level. Consequently, SCCs are used in a variety of differently calculated DBVs. In addition to the *average SCC* of different lactation groups and the *proportion of cows with SCC > 400,000 cells per millilitre*, variables describing *udder infection*—defined by an SCC of more than 150,000 cells per millilitre in primiparous and 250,000 cells per millilitre in multiparous cows—were investigated. In some cases, an additional temporal component was considered, and thus chronically infected cows were monitored. Again, although several variables were tested, none of them overlapped between studies by different authors. Because impaired herd udder health status can manifest itself in both higher average SCCs and higher proportions of udder-infected cows, the different variables are closely related and can be evaluated together. *Mastitis treatment incidence*, which may be increased in herds with poor udder health, was also included. Univariate associations with the ABMs hindquarter cleanliness, lean body condition, lesions or swellings, frequency of displacements, and lameness were demonstrated for several udder health variables in one study [28] (Table 7).

The multivariable analyses revealed positive associations between the DBVs *average SCC of cows 0–60 DIM* [28] as well as *new udder infections (%)* [27] and the percentage of cows with dirty hindquarters. The variables *udder infection (%)* [28] and *average SCC of cows 0–60 DIM* [27,28] were both positively related to the proportion of very lean cows. Herds with a higher proportion of very lean cows also had a higher *incidence of mastitis treatments* and a higher *incidence risk of udder infections* [25]. *Heifer udder infections (%)* [27,28] and *average SCC of cows 120–210 DIM* [28] both showed positive associations with the proportion of cows with lesions or swellings. Although three udder health variables were associated with the proportion of severely lame cows in a univariable analysis [28], the multivariable analysis only showed a positive association with the prevalence of moderately lame cows for the *average SCC of cows 120–210 DIM* [27]. Similarly, for the frequency of displacements—a measurement of social behaviour— only one multivariate association was shown with the *average SCC of cows 0–60 DIM* [27], although several DBVs were associated in univariable analyses [28].

Overall, the reported associations between DBVs related to udder health and ABMs were positive, which means that herds with better udder health status also performed better with regard to the ABMs. In contrast, the mean time needed to lie down (according to WQ the time from bending the carpal joint to complete lying on the lying surface) was increased in herds with more *new infections*, but decreased in herds with a higher rate of *heifer udder infections* [27].

#### 5.2.5. Bulk Milk Somatic Cell Count

In addition to the individual-cow SCC, bulk milk somatic cell counts (BMSCCs) are also used to monitor udder health, because these are usually available for all milk-delivering farms. All five studies included in this section evaluated the BMSCCs (Table 8). As with the animal-specific SCC, the *BMSCC* was repeatedly associated with the occurrence of severely lame cows. For the *BMSCC over 90 days* before the lameness assessment, a negative association with the proportion of severely lame cows was described [31]. In contrast, another study showed that the *mean BMSCC* on farms with less than 16% severely lame cows was lower than the *mean BMSCC* on farms with a higher lameness prevalence [29]. Associations between BMSCC and other ABMs could not repeatedly be demonstrated in other studies.

#### 5.2.6. Milk Yield

Data on the milk yield of individual cows were used as DBVs at farm level in four of the studies. However, different calculations were used: *mean milk yield per cow and day*, *energy-corrected milk yield (ECM) per cow year*, and *average 305-day milk yield*. In addition, the variation in milk yield expressed as SD or coefficient of variation (CV) was used as DBV. Because all authors calculated the variables either for the whole herd or for the same parity groups, the identified associations for these DBVs were easier to compare than, for example, DBVs related to udder health. The studies of de Vries et al. [27,28] and Otten et al. [31] revealed associations of DBVs based on the milk yield to various ABMs (Table 9). However, another study [25] showed no associations between ABMs and the *average 305-day milk yield* in any parity group considered. *Average milk yield per cow and day (kg)* [27] and *ECM per cow year (90 days)* [31], each considering cows of all parities, were both negatively associated with the proportion of cows with dirty hindquarters. By contrast, the *ECM of cows of third and higher lactation (90 days)* was associated positively with this ABM [31]. For the proportion of cows with dirty udders, a nonlinear association with the *average milk yield per cow and day (kg)* [27] and a negative association with the *ECM per cow year (90 days)* [31] could be demonstrated. The *average milk yield per cow and day (kg)* showed a univariable association with the prevalence of severely lame cows [28], and the variable *ECM per cow year (90 days)* was negatively associated with this ABM [31]. Moreover, the *ECM of first lactating cows* was associated with the proportion of severely lame cows, but the direction of this association was not consistent for two different calculation periods [31].

For the proportion of cows with dirty udders, a nonlinear association with *average milk yield per cow and day (kg)* [27] and a negative association *with ECM per cow year (90 days)* [31] could be shown. All other demonstrated associations between DBVs addressing milk yield and ABMs could not repeatedly be demonstrated in other publications.

#### 5.2.7. Average Days in Milk

DBVs based on DIM were exclusively studied by de Vries et al. [27,28] (Table 10). Between *average DIM* and the proportion of severely lame cows, nonlinear relationships were found [27,28]. A nonlinear [27] and a nonlinear negative [28] relationship were found to the proportion of cows with lesions or swellings.

#### 5.2.8. Milk Constituents

For the milk constituents milk fat and milk protein, as well as the fat-to-protein ratio, associations between DBVs and ABMs were identified in the studies by de Vries et al. [27,28] (Table 11). Two studies showed a nonlinear association between the *milk fat content* and the proportion of very lean cows [27,28]. The *average ratio fat/protein of cows 0–60 DIM* showed a nonlinear [27] and a nonlinear negative [28] relationship to the percentage of cows with lesions and swellings. Furthermore, an increased *average ratio fat/protein of cows 0–60 DIM* was associated with an increased proportion of severely lame cows [27,28].

For milk urea content, one further study also provided associations with ABMs. The *prevalence of cows with urea remarks* [25] was negatively associated with the proportion of cows with injuries or inflammation and the *average milk urea content* [27] was negatively associated with the proportion of cows with lesions or swellings. Associations with other ABMs could not be repeatedly demonstrated in the identified publications.

#### 5.2.9. Fertility

Although DBVs related to fertility were included in only four studies, many variables were investigated (Table 12), with many of these linked to the calving interval. The calving interval itself is composed of the interval from calving to first insemination service (CFSI), the period from first to successful insemination, and the gestation period, which is usually 283 days for dairy cows. Delays in CFSI or a prolonged insemination period—e.g., due to several necessary inseminations and thus a low non-return rate in 56 days—can thus prolong the calving interval. The variation of the calving interval, expressed as CV, was also investigated.

Many DBVs related to fertility were associated with the proportion of lean cows. For five of these DBVs, univariable associations at the significance level α = 0.2 could be detected. A nonlinear [28] and a positive association [25] with the proportion of lean cows was found for the *average calving interval*. The *CV calving interval* [25] and the *average expected calving interval* [27] were negatively associated with this ABM. Moreover, the *CFSI* showed a nonlinear relationship with the proportion of lean cows [27], and a higher level of *average services per cow* was positively associated with this ABM [27]. Finally, the *non-return rate 56 days* was found to be higher on farms with ≥7% of the cows classified as very lean. This result implies that farms with poor welfare [28] had better first insemination success.

Furthermore, several DBVs related to fertility were linked to lameness prevalence. Both *realised* and *expected average calving interval* were univariably associated with the proportion of severely lame cows [28], and the *proportion of cows with late ongoing artificial inseminations* was positively associated with this ABM [25]. The prevalence of moderately lame cows was also negatively related to the *average services per cow* and nonlinearly related to *CFSI* [27].

Two studies revealed links between DBVs related to fertility and integument alterations. *Average services per cow* and the *proportion of cows with >2 services* were associated with the percentage of cows with lesions or swellings [28]. The *average calving interval* and the *CV calving interval* were both negatively related to the percentage of cows with injuries or inflammation [25].

#### 5.2.10. Data-Based Variables without Clear Patterns in the Associations with Animal-Based Measurements

For some DBVs, associations with ABMs were neither repeatedly shown, nor did clear patterns emerge. These include DBVs based on cow age, herd disease or biosecurity status, and net milk yield. Furthermore, the DBVs that included culls and their reasons as well as veterinary treatments and diagnoses did not show clear patterns in their associations to ABMs. For slaughterhouse data, often considered a promising option for implementing animal welfare monitoring based on DBVs, the associations were partially repeated for different calculation periods within one study [31] but not confirmed by other studies. An extensive list of all associations between DBVs and ABMs that were not repeatedly shown to date can be found in the Appendix A.

### 5.3. Discussion of the Associations between Data-Based Variables and Animal-Based Measurements and the Resulting Implications for Animal Welfare Monitoring

All five papers reviewed in this section showed significant associations between DBVs and specific ABMs. Among the DBVs tested, those based on cow mortality were particularly frequently related to different ABMs and were therefore considered promising DBVs. Since the studies used DBVs based on similar data, the results could be compiled to determine whether associations between ABMs and DBVs are repeated across the different studies and thus confirmed. In this process, it was observed that the DBVs varied greatly with regard to their definitions as well as the age or production groups considered, so that many different variants were examined. Only two studies conducted by the same authors [27,28] used identical variants of DBVs. In addition, the comparison of DBVs was partly complicated by a lack of precise definitions of the DBVs. For instance, with variables based on calf mortality, it was partly not further defined up to which age calves were considered for the DBVs. For future work, we therefore suggest including detailed definitions or calculation methods of the DBVs, as has already been done in a study by Krug et al. [30]. For example, in this study the mortality rate was calculated as ‘incidence of on-farm deaths and emergency slaughter reported in death/100 animal-year at risk’. Together with the information on the calculation of animal years at risk, the publication provides a clear indication of the cows considered for the calculation of the mortality rate.

Repeated associations between specific DBVs and ABMs were most frequently provided by the two studies by de Vries et al. [27,28] that used identical variants of DBVs. Despite the wide variety of DBVs and the partly lacking definitions, some associations were confirmed between different studies and may thus indicate a universal association. The largest amount of confirmation was found for the association between DBVs based on increased cow mortality with an increased proportion of severely lame cows [27,28,29]. As increased mortality has additionally been reported to be a predictor of herds with a higher proportion of severely lame cows [29], these DBVs could potentially be used as an indicator. Beyond this, confirmations of associations between specific DBVs and ABMs were rare. However, when the associations of related DBVs were evaluated together, patterns in the associations emerged more often. Although the variants of the DBVs make the comparison of the studies difficult, other factors may also lead to the overall rather limited confirmations of associations between the studies.

Firstly, some ABMs, such as the frequency of head butts or qualitative behavioural assessments were each investigated in only one study. Therefore, it is not possible to obtain repeated associations for these ABMs. To use DBVs as indicators of multidimensional animal welfare, all areas of animal welfare must be covered. Future research should therefore also evaluate ABMs of social behavior and emotional state.

Secondly, the ABMs that were applied were also partly comparable to a limited extent only. Regarding integument alterations, one survey assessed extensive lesions and swellings [25], whereas the WQ protocol distinguished between the proportion of cows with lesions or swellings and the proportion of cows with hairless areas. Otten et al. [31] instead used a score for the integumentary alterations of a body region—e.g., tarsus, legs, or body—that combined hairless areas, lesions, and swellings in that region. In this context, if the study design and aim allow it, established and widely used ABMs could be used.

In addition to the previously mentioned reasons, another explanation for the rare confirmations of the associations could be that the studies calculated DBVs for different time periods. Otten et al. [31] determined the DBVs for periods covering 365, 180, and 90 days preceding the on-farm assessment and reported that most of the associations found were not constant over the periods. In addition, one should consider that the analyses did not aim to identify causal relationships, and thus the influence of confounding factors cannot be excluded.

## 6. Relationships between Data-Based Variables and Welfare Quality Criteria and Principles, CReNBA Areas, and Overall Scores

In this section, all identified papers are reviewed that examine the relationship of DBVs with WQ criteria and principles, CReNBA areas, and overall scores (see Box 1) or that aim to predict farm-level welfare expressed in these scores. Three of the 13 identified publications examined one or a few related DBVs for their association with dairy herd welfare [32,33,34].

Coignard et al. [34] investigated the relationship between milk yield and dairy welfare at farm level. For 125 French dairy farms whose animal welfare status was surveyed with the WQ protocol, individual-cow milk yields were determined for 30 days before and after the on-farm assessment. The associations of the milk yield with the WQ criteria, the WQ principles, and the WQ overall score were analysed using linear mixed models. At the criteria level, the occurrence of agonistic behaviour and a poor emotional state, i.e., lower values in the Qualitative Behaviour assessment of the WQ, were associated with lower milk yield. Herds that scored worse in the criteria ‘absence of injury’ and ‘absence of disease’ had a higher milk yield. The principle ‘good health’, which is made up of these two criteria, was thus negatively linked to milk yield. However, no significant association of milk yield to the overall score could be shown.

The relationship between dairy herd welfare and the reproductive parameters *calving rate* and *CFSI* was studied by Grimard et al. [33]. The welfare of 124 dairy herds was assessed using the WQ protocol, and the relationship with the two parameters was checked at the level of WQ criteria, WQ principles, and WQ overall score. The overall score showed a significant association with *CFSI*: the higher the overall score was, the shorter was the *CFSI*. The *calving rate*, in contrast, was not linked to overall welfare but showed a positive relationship to the principle ‘good housing’.

Ginestreti et al. [32] studied whether herd welfare status could be predicted using the bulk milk parameters *BMSCC, total bacterial count,* and *fat, protein,* and *urea contents*. For 287 farms, the results of routine on-farm welfare assessments with the CReNBA protocol were obtained from the Italian animal welfare database. This welfare assessment provided an overall score consisting of three areas: management and stock training, housing, and ABMs. The examination of the relationships between the bulk milk parameters and overall welfare or the area scores did not reveal a significant relationship between animal welfare and *milk fat content*. All other DBVs showed only weak relationships. Consequently, the authors assigned a very limited predictive value to data gained by bulk milk analysis.

Finally, the objective of four studies [25,26,30,31], including two already mentioned in Section 5 [25,31], was to form sets of DBVs that could predict the farm-level welfare status and thus be suitable as predictive models. In a study by Sandgren et al. [25], 13 of 55 visited herds were considered to have poor animal welfare, because they scored among the worst 10% in at least two of nine assessed ABMs. Eighteen DBVs were found to be associated with these ABMs in a multivariable analysis. By means of a systematic selection process, the authors recognised three DBVs that, in combination, were suitable for identifying herds with poor animal welfare status: *cows with late ongoing artificial insemination*, *late-bred heifers,* and *calf mortality*. This predictive model correctly classified 77% of the herds with poor welfare, with a sensitivity of 62%. Three further models, including *cow mortality* and *young stock mortality*, classified 76% of these herds correctly, with a sensitivity of 77%.

Based on the aforementioned approach, Nyman et al. [26] investigated whether a set of DBVs could reliably identify dairy herds with good welfare. Among the 55 visited herds, 28 were classified as having good welfare, because they were not among the worst 10% of farms in any of the nine selected ABMs. Statistical analysis yielded six DBVs that together correctly classified 96% of the herds with good welfare: *cows with late ongoing artificial insemination, late-bred heifers, cow mortality, stillbirth rate, mastitis incidence*, and *incidence of feed-related diseases*. Nyman et al. [26] additionally suggested combining the developed predictive model with the model of Sandgren et al. [25] to allow a refined classification into farms with presumed good welfare, farms with presumed poor welfare, and farms that could not be classified.

Krug et al. [30] used DBVs from the Portuguese database for bovine identification and registration to build a model to detect dairy herds with poor welfare. The welfare status of 24 herds was assessed using the WQ protocol, resulting in five herds having poor welfare according to the WQ overall score. Of the 15 DBVs examined, the *proportion of on-farm deaths* and the *ratio of female to male births* differed significantly between herds with good and poor welfare status in the univariable analysis. In addition, data mining was used to detect the best performing group of DBVs for the detection of herds with poor animal welfare status. This resulted in a classification tree model with a sensitivity of 70.0% and specificity of 78.9%, based on the variables *on-farm deaths* and *proportion of calving intervals >430 days*.

Otten et al. [31] aimed to use DBVs to predict the animal welfare status at farm level by means of an animal-based index. The index consisted of 12 aggregated, weighted ABMs and was applied on 73 farms. Similarly, 21 DBVs were aggregated linearly into a weighted data-based index, considering three time periods (covering 90, 180, and 365 days before the farm visit). Of these three calculation periods, only the 180-day data-based index was significantly related to the animal-based index, providing only poor model fit and predictive value. It was concluded that DBVs can give a first indication of the welfare status of a farm. However, to assess the real welfare status, the authors still considered an on-farm welfare assessment to be necessary.

In summary, some DBVs, including milk yield and the fertility parameter *CFSI*, showed associations with WQ criteria, WQ principles, or the WQ overall score. For the bulk milk parameters, on the other hand, only weak associations with the animal welfare areas of the CReNBA protocol were found. One should consider that criteria, principles, and overall scores are based on measurements that are aggregated and weighted according to the animal welfare definitions of the protocol used and partially compensate each other. Thus, the identified associations are dependent on the given welfare definition and might not be confirmed if a different definition of welfare were applied. However, because these assessments can only provide an approximation of the actual animal welfare status, the associations must be interpreted with caution. With regard to the prediction of case herds—i.e., herds with poor or good animal welfare—three of the studies [25,26,30] provided predictive models formed by DBVs that achieved sufficient sensitivities and specificities to give a first impression of farm-level welfare status. In contrast, one publication [31] showed only limited correlations between a data-based index and an animal-based welfare assessment. It should be noted that the predictive models were built using statistical methods, applying selection processes that included those DBVs in the model that best identified the case herds of the study populations. The predictive sets in the presented studies were not applied to additional data sets and consequently have not been validated. This makes the models highly dependent on the conditions under which they were designed, i.e., the influencing factors and dairy farms used as the study population. Nevertheless, the authors of the presented studies considered DBVs as useful tools to give first evidence on the animal welfare status of farms. However, to obtain the actual welfare status, the results of the predictive models would need to be confirmed by comprehensive on-farm surveys.

## 7. Conclusions and Implications for Future Animal Welfare Monitoring

In summary, we could identify relatively few studies focusing on the relationships between DBVs and dairy welfare at farm level. In the 13 studies reviewed here, the DBVs investigated were similar with regard to the raw data collected, but differed in the definition and survey of animal welfare. To describe the farm-level welfare status, either the presence of welfare violations was considered or the welfare of dairy cows was surveyed at the level of ABMs, welfare overall scores, or scores for areas of multidimensional welfare definitions. Three studies that investigated the suitability of DBVs to detect cattle farms violating animal welfare suggested that DBVs may have the potential to contribute to animal welfare monitoring. The studies examining sets of DBVs to predict dairy farms with good or poor welfare status mostly showed the suitability of DBVs for this purpose. Nevertheless, comprehensive on-farm surveys are necessary to determine the actual animal welfare status. In addition, several DBVs were related to scores of welfare assessments such as WQ criteria, CReNBA areas, and overall score. The evaluation of relationships between DBVs and specific ABMs, such as lameness or body condition, yielded a large number of associations. In this context, DBVs based on mortality were particularly frequently associated with different ABMs. Owing to varying calculations and the consideration of different age or performance groups, a large number of DBV variants were examined. Together with sometimes missing definitions of the variables used, this led to a limited comparability of the studies. This may be a reason why repeated associations of specific DBVs with ABMs were rare.

Overall, the literature included in this review indicates a wide range of potential applications for DBVs. However, the limited number of studies and lack of validation of DBVs necessitate further research to fully assess the value of DBVs for animal welfare monitoring. To account for the multidimensional nature of animal welfare, comprehensive sets of ABMs should be used. In addition, the use of validated and widely accepted ABMs could increase comparability between studies. Future work may be based on the variables examined so far, using comparable DBVs if possible, and include information on their definitions and calculation. The investigation of, as yet, rarely used data sources, such as reports from slaughter examinations, could provide additional DBVs with the potential for monitoring the welfare of dairy cows.

## Figures and Tables

**Table 1 animals-11-03458-t001:** Search terms used in the initial literature identification.

Search Context	Search Term
Synonyms for dairy cows	cattle, bovine, ruminant, dairy, dairying
Terms related to data, welfare, and health	routinely collected data, routine herd data, census data, pre-collected data, national database, register-based, welfare, welfare assessment, welfare quality, well-being, wellbeing, animal-based, health assessment, health monitoring, monitoring, surveillance, health indicator, movement, transport, mortality, carcass, carcass condemnation

**Table 2 animals-11-03458-t002:** Overview of 13 studies identified with the literature research.

Reference	Aim	Welfare Definition	Welfare Assessment on Farm	Analysis
[22]	Identification of DBVs to identify herds with animal welfare violations	Poor welfare defined as farms with animal welfare violations according to official welfare inspectors	None. Selection of target herds with confirmed animal welfare violations	Analysis of DBVs from herds with animal welfare violations
[23]	Validation of DBVs as predictors of herds with animal welfare violations	See above	See above	Comparison of the distribution of possible key indicators in 18 case herds and the national herd
[24]	Prediction of herds with animal welfare violations by use of DBVs	Violations of animal welfare defined as: presence of sick animals without separation in sick pen or presence of animals whose condition would require euthanasia	Examination of the occurrence of at least 1 of the 2 most common violations of animal welfare	Associations between 25 DBVs and violations of animal welfare
[25]	Predicting herds with poor animal welfare	‘Poor animal welfare’ defined as: scoring among the 10% of the worst farms in at least 2 of the 9 ABMs	Welfare assessment consisting of 9 selected ABMs (5 surveyed exclusively for dairy cows, 3 surveyed for dairy cows, young stock and calves)	Associations between 65 DBVs and single ABMs; selection of sets of DBVs to detect herds of poor welfare status
[26]	Predicting herds of good welfare status	‘Good animal welfare’ defined as: no score among the 10% of the worst in any of the 9 ABMs	Data collected by [25]	Based on associations detected by [25]
[27]	Prediction of herd welfare status by using DBVs	Herd welfare definition split into the WQ measurements	WQ protocol for dairy cows	Associations between 46 DBVs and 22 WQ measurements; prediction models for 22 measurements of the WQ
[28]	Identification of herds with poor welfare status by using DBVs	Herd welfare definition split into the WQ measurements	Six selected ABMs of the WQ protocol	Associations between 41 DBVs and 6 ABMs of the WQ protocol; prediction models for 6 WQ measurements
[29]	Identification of herds with high lameness prevalence by using DBVs	High lameness prevalence as part of impaired welfare	Lameness scoring	Associations between continuous and dichotomised DBVs and high lameness prevalence
[30]	Detecting herds with poor welfare status	Poor welfare status: WQ overall score ‘not classified’	WQ protocol for dairy cows	Associations between 15 DBVs and the WQ overall score ‘not classified’; decision tree model to predict herds with poor welfare status
[31]	Validation of a data-based and a resource-based welfare index against an animal-based welfare index	Welfare definition consisting of 12 weighted ABMs	Twelve ABMs	Associations between 24 DBVs and single ABMs; validation of the data-based index against the animal-based index
[32]	Prediction of herd welfare status by bulk milk analysis data	According to the CReNBA welfare assessment protocol	Surveys allocated to the 3 areas: farm management and training, housing, and ABMs.	Association of bulk milk parameters and the CReNBA area scores as well as overall score
[33]	Associations between 2 fertility variables and herd welfare	According to WQ	WQ protocol for dairy cows	Associations between 2 fertility variables and 11 WQ criteria, 4 WQ principles, and the WQ overall score
[34]	Association between milk yield and dairy herd welfare status	According to WQ	WQ protocol for dairy cows	Associations between milk yield data and 11 WQ criteria, 4 WQ principles, and the WQ overall score

Abbreviations: DBV = data-based variable, ABM = animal-based measurement, WQ = Welfare Quality, CReNBA = Centro di Referenza Nazionale per il Benessere Animale.

**Table 3 animals-11-03458-t003:** Data sources used for the calculation of data-based variables used as potential predictors of dairy herd welfare in 13 identified studies.

Reference	[25,26]	[30]	[27]	[28]	[31]	[24]	[22]	[23]	[34]	[33]	[32]	[29]
Identification and registration	x	x	x	x	x	x	x	x				x
Bulk milk data	x		x	x	x	x					x	x
Milk yields	x		x	x	x	x			x			
Slaughterhouse data	x	x			x	x						x
Reproduction data	x	x	x	x	x					x		
Individual somatic cell counts	x		x	x								
Milk contents (individual)	x		x	x								
Disease and treatment data	x				x	x						
Disease status			x	x								
Reasons for culling	x											

x indicates the use of the data sources for the respective work.

**Table 4 animals-11-03458-t004:** Associations between data-based variables (DBVs) related to cow mortality and animal-based measurements (ABMs) extracted from five scientific publications.

DBV	ABM	Type ^1^ and Direction ^2^	Reference
Cow mortality, continuous	Severely lame cows	( )	[29]
On-farm mortality of cattle 0–60 DIM (%)		{+}	[27]
On-farm mortality of cattle 120–210 DIM (%)		{+}	[27]
On-farm mortality of cattle aged >2 years (%)		( )	[28]
On-farm mortality of cattle aged >2 years (%)	Severely lame cows >11.9%	{+}	[28]
On-farm mortality of cattle >210 DIM (%)	Moderately lame cows	{+}	[27]
On-farm mortality of cattle 120–210 DIM (%)	Cows with hairless patches	{−}	[27]
On-farm mortality of cattle >210 DIM (%)	Cows with lesions or swellings	{+}	[27]
On-farm mortality of cattle aged >2 years (%)		( )	[28]
On-farm mortality of cattle 0–60 DIM (%)	Lean or very lean cows	{+}	[27]
On-farm mortality of cattle 120–210 DIM (%)		{−}	[27]
Cow mortality	Lean young stock	{+}	[25]
Survival early lactation (3 months)		{+}	[25]
On-farm mortality of cattle 0–60 DIM (%)	Collisions with stall components	{+}	[27]
On-farm mortality of cattle 120–210 DIM (%)		{+}	[27]
On-farm mortality of cattle >210 DIM (%)	Time needed to lie down (mean)	{+}	[27]
On-farm mortality of cattle 60–120 DIM (%)		{−}	[27]
On-farm mortality of cattle 120–210 DIM (%)	Frequency of displacements	{−}	[27]
On-farm mortality of cattle 0–60 DIM (%)	Frequency of head butts	{+}	[27]
Cow mortality (90 days)	Avoidance distance, shyness	(−)	[31]
On-farm mortality of cattle 0–60 DIM (%)	Dirty udder	{−}	[27]
On-farm mortality of cattle 0–60 DIM (%)	Number or length of drinkers	{+}	[27]
Cow mortality (365 days)	Cows with impaired hair coat	(−)	[31]
On-farm mortality of cattle 0–60 DIM (%)	Cows with nasal discharge	{+}	[27]
On-farm mortality of cattle 0–60 DIM (%)	Cows with vulvar discharge	{+}	[27]
Survival early lactation (3 months), primiparous cows	Rising abnormal or impaired	{−}	[25]

Abbreviations: DIM = days in milk. ^1^ Type of associations indicated by ( ) for univariable associations and { } for multivariable associations. ^2^ Direction of associations indicated by + for positive associations and − for negative associations, empty brackets indicate that no direction was given.

**Table 5 animals-11-03458-t005:** Associations between data-based variables (DBVs) related to calf mortality and animal-based measurements (ABMs) extracted from four scientific publications.

DBV	ABM	Type ^1^ and Direction ^2^	Reference
Calf mortality (90 days)	Lean or very lean cows	(+)	[31]
On-farm mortality of cattle aged 0–3 days (%)		( )	[28]
Calf mortality 2–6 months		(+)	[25]
Stillbirth (%)	Moderately lame cows	{+}	[27]
Calf mortality (90 days)	Severely lame cows	(+)	[31]
On-farm mortality of cattle aged 0–3 days (%)		( )	[28]
Calf mortality 0–24 h	Rising abnormal or impaired	{+}	[25]
Calf mortality 1–90 days		{+}	[25]
Calf mortality 1–90 days	Dirty cows	(+)	[25]
On-farm mortality of cattle aged 0–3 days (%)	Cows with dirty hindquarter	( )	[28]
Heifer mortality (90 days)	Avoidance distance, shyness	(−)	[31]
On-farm mortality of cattle aged 4–365 days (%)	Dystocia %	{−}	[27]
Calf mortality (90 days)	Integument alterations tarsus	(+)	[31]
Calf mortality 2–6 months	Lean young stock	(+)	[25]
Calf mortality 2–6 months	Lean calves	{+}	[25]

^1^ Type of associations indicated by ( ) for univariable associations and { } for multivariable associations. ^2^ Direction of associations indicated by + for positive associations, − for negative associations, empty brackets indicate that no direction was given.

**Table 6 animals-11-03458-t006:** Associations between data-based variables (DBVs) related to herd size and animal-based measurements (ABMs) extracted from two scientific publications.

DBV	ABM	Type ^1^ and Direction ^2^	Reference
Herd size (number of lactating cows)	Lean or very lean cows	( )	[28]
	Lean or very lean cows >7.0%	{+}	[28]
Herd size (number of lactating cows)	Cows with dirty hindquarter	( )	[28]
	Cows with dirty hindquarter >60.4%	{+}	[28]
Herd size (number of lactating cows)	Severely lame cows	( )	[28]
Herd size (number of cows)		{+}	[27]
Herd size (number of lactating cows)	Avoidance distance	( )	[28]
Herd size (number of cows)	≥2 drinkers available	{−}	[27]
Change in herd size (%)	Severely lame cows	( )	[28]
	Severely lame cows >11.9%	{−}	[28]
Change in herd size (%)	Lean or very lean cows	( )	[28]
Change in herd size (%)	Lying outside lying area	{−}	[27]

^1^ Type of associations indicated by ( ) for univariable associations and { } for multivariable associations. ^2^ Direction of associations indicated by + for positive associations, − for negative associations, empty brackets indicate that no direction was given.

**Table 7 animals-11-03458-t007:** Associations between data-based variables (DBVs) related to udder health and animal-based measurements (ABMs) extracted from three scientific publications.

DBV	ABM	Type ^1^ and Direction ^2^	Reference
Average SCC of cows >210 DIM	Cows with dirty hindquarter	( )	[28]
Average SCC of cows 0–60 DIM		( )	[28]
Average SCC of cows 60–120 DIM		( )	[28]
Cows with SCC > 400,000 cells/mL (%)		( )	[28]
New udder infection (%)		{+}	[27]
Udder infection (%)		( )	[28]
Average SCC of cows 0–60 DIM	Cows with dirty hindquarter >60.4%	{+}	[28]
Average SCC of cows 0–60 DIM	Lean or very lean cows	{+}	[27]
Average SCC of cows 0–60 DIM		( )	[28]
Cows with SCC > 400,000 cells/mL (%)		( )	[28]
Incidence of mastitis treatment		{+}	[25]
Incidence risk of udder infections (%)		(+)	[25]
New udder infection (%)		( )	[28]
Udder infection (%)		( )	[28]
Udder infection (%)	Lean or very lean cows >7.0%	{+}	[28]
Average SCC of cows 120–210 DIM	Cows with lesions or swellings	( )	[28]
Heifer udder infection (%)		{+}	[27]
Heifer udder infection (%)		( )	[28]
New udder infection (%)		( )	[28]
Average SCC of cows 120–210 DIM	Cows with lesions or swellings >61.2%	{+}	[28]
Heifer udder infection (%)		{+}	[28]
Average SCC of cows 0–60 DIM	Frequency of displacements	{+}	[27]
Average SCC of cows 0–60 DIM		( )	[28]
Cows with SCC > 400,000 cells/mL (%)		( )	[28]
Udder infection (%)		( )	[28]
Average SCC of cows 120–210 DIM	Moderately lame cows	{+}	[27]
Cows with SCC > 400,000 cells/mL (%)	Severely lame cows	( )	[28]
New udder infection (%)		( )	[28]
Udder infection (%)		( )	[28]
Heifer udder infection (%)	Mean time needed to lie down	{−}	[27]
New udder infection (%)		{+}	[27]
Udder infection (%)	Dehorning of calves (yes)	{−}	[27]
Average SCC of cows 120–210 DIM	Qualitative behaviour assessment	{+}	[27]
Prevalence of cows with chronically high SCC	Dirty calves	{+}	[25]
Heifer udder infection (%)	Cows with dirty udder	{+}	[27]

Abbreviations: DIM = days in milk, SCC = somatic cell count. ^1^ Type of associations indicated by ( ) for univariable associations and { } for multivariable associations. ^2^ Direction of associations indicated by + for positive associations, − for negative associations, empty brackets indicate that no direction was given.

**Table 8 animals-11-03458-t008:** Associations between data-based variables (DBVs) gained by bulk milk analysis and animal-based measurements (ABMs) extracted from five scientific publications.

DBV	ABM	Type ^1^ and Direction ^2^	Reference
BMSCC (90days)	Severely lame cows	(−)	[31]
BMSCC, continuous		( )	[29]
BMSCC (180 days)	Rising abnormal or impaired	(−)	[31]
BMSCC (90 days)		(+)	[31]
BMSCC	Frequency of displacements	( )	[28]
	Lean or very lean cows	(+)	[25]
	Dirty calves	(+)	[25]
	Cows with dirty hindquarter	( )	[28]
	Cows with lesions or swellings	( )	[28]
	Cows lying outside lying area	{+}	[27]
Butyric acid bacteria (yes)	Frequency of head butts	{+}	[27]

Abbreviations: BMSCC = bulk milk somatic cell count. ^1^ Type of associations indicated by ( ) for univariable associations and { } for multivariable associations. ^2^ Direction of associations indicated by + for positive associations, − for negative associations, empty brackets indicate that no direction was given.

**Table 9 animals-11-03458-t009:** Associations between data-based variables (DBVs) addressing milk yield and animal-based measurements (ABMs) extracted from four scientific publications, grouped by parity groups considered.

DBV	ABM	Type ^1^ and Direction ^2^	Reference
DBVs Considering the Whole Herd			
Average milk yield per cow and day (kg)	Cows with dirty hindquarter	( )	[28]
Average milk yield per cow and day (kg)		{−}	[27]
ECM per cow year (90 days)		(−)	[31]
Average milk yield per cow and day (kg)	Cows with dirty udder	{~}	[27]
ECM per cow year (90 days)		(−)	[31]
Average milk yield per cow and day (kg)	Severely lame cows	( )	[28]
ECM per cow year (90 days)		(−)	[31]
Average milk yield per cow and day (kg)	Frequency of displacements	( )	[28]
Average milk yield per cow and day (kg)	Frequency of displacements >0.58 per cow and hour	{+}	[28]
ECM per cow year (365 days)	Impaired claw conformation	(−)	[31]
ECM per cow year (90 days)		(−)	[31]
ECM per cow year (90 days)	Rising abnormal or impaired	(−)	[31]
ECM per cow year (90 days)	Impaired hair coat	(−)	[31]
ECM per cow year (90 days)	Avoidance distance, shyness	(+)	[31]
ECM per cow year (90 days)	Cows with dirty legs	(−)	[31]
DBVs Considering 1st Lactation Cows			
ECM 1st lactation cows (365 days)	Severely lame cows	(−)	[31]
ECM 1st lactation cows (90 days)		(+)	[31]
ECM 1st lactation cows (90 days)	Impaired hair coat	(+)	[31]
DBVs Considering 2nd and Higher Lactation Cows			
ECM 2nd lactation cows (365 days)	Rising abnormal or impaired	(−)	[31]
ECM 2nd lactation cows (365 days)	Impaired claw conformation	(−)	[31]
ECM 2nd lactation cows (180 days)	Impaired hair coat	(+)	[31]
ECM 2nd lactation cows (365 days)	Lean or very lean cows	(−)	[31]
ECM 3rd and higher lactation cows (90 days)	Cows with dirty hindquarter	(+)	[31]
ECM 3rd and higher lactation cows (180 days)	Integument alterations tarsus	(−)	[31]
DBVs Considering Deviation of Milk Yields			
SD ECM 1st lactation cows (90 days)	Integument alterations body	(+)	[31]
	Integument alterations carpus	(+)	[31]
SD ECM 1st lactation cows (180 days)	Impaired claw conformation	(+)	[31]
SD ECM 1st lactation cows (90 days)		(+)	[31]
SD ECM 1st lactation cows (180 days)	Avoidance distance, shyness	(+)	[31]
CV average milk yield per cow and day (kg) 2nd to 3rd lactation month, primiparous cows	Dirty young stock	{+}	[25]
	Severely lame cows	{+}	[25]
SD ECM 3rd and higher lactation cows (90 days)	Avoidance distance, shyness	(−)	[31]
	Cows with dirty hindquarter	(+)	[31]

Abbreviations: ECM = energy-corrected milk yield, CV = coefficient of variation. ^1^ Type of associations indicated by ( ) for univariable associations and { } for multivariable associations. ^2^ Direction of associations indicated by + for positive associations, − for negative associations, and ~ for nonlinear associations, empty brackets indicate that no direction was given.

**Table 10 animals-11-03458-t010:** Associations between data-based variables (DBVs) based on days in milk and animal-based measurements (ABMs) extracted from two scientific publications.

DBV	ABM	Type ^1^ and Direction ^2^	Reference
Average DIM	Severely lame cows	{~}	[27]
	Severely lame cows	( )	[28]
	Severely lame cows >11.9%	{~}	[28]
	Cows with lesions or swellings	{~}	[27]
	Cows with lesions or swellings	( )	[28]
	Cows with lesions or swellings >61.2%	{~, −}	[28]
	Cows with dirty legs	{+}	[27]
Cows 0–60 DIM (%)	Cows with nasal discharge	{−}	[27]

Abbreviations: DIM = days in milk. ^1^ Type of associations indicated by ( ) for univariable associations and { } for multivariable associations. ^2^ Direction of associations indicated by + for positive associations, − for negative associations, and ~ for nonlinear associations, empty brackets indicate that no direction was given.

**Table 11 animals-11-03458-t011:** Associations between data-based variables (DBVs) based on milk contents and animal-based measurements (ABMs) extracted from three scientific publications.

DBV	ABM	Type ^1^ and Direction ^2^	Reference
Average milk fat (%)	Lean or very lean cows	{~}	[27]
	Lean or very lean cows	( )	[28]
	Lean or very lean cows >7.0%	{~}	[28]
	Frequency of displacements	( )	[28]
	Frequency of head butts	{+}	[27]
	Cows lying outside lying area	{~}	[27]
	Dehorning young stock (yes)	{+}	[27]
	Qualitative behaviour assessment	{~}	[27]
Average milk protein (%)	Cows with diarrhoea	(−)	[27]
	Cows with dirty udder	{+}	[27]
	Time needed to lie down	{+}	[27]
Average ratio fat/protein 0–60 DIM	Cows with lesions or swellings	{~}	[27]
	Cows with lesions or swellings	( )	[28]
	Cows with lesions or swellings >61.2%	{~, −}	[28]
	Severely lame cows	{+}	[27]
	Severely lame cows	( )	[28]
	Severely lame cows >11.9%	{+}	[28]
	Lean or very lean cows	( )	[28]
	Cows with dirty udder	{+}	[27]
	Qualitative behaviour assessment	{~}	[27]
Prevalence of cows with high levels of urea	Lean or very lean cows	(+)	[25]
	Lean young stock	(+)	[25]
Prevalence of cows with low levels of urea	Lean calves	(−)	[25]
	Dirty calves	(−)	[25]
Prevalence of cows with urea remarks (high and low)	Lean or very lean cows	{+}	[25]
	Cows with injuries or inflammation	(−)	[25]
Average milk urea (%)	Cows with lesions or swellings	(−)	[27]
	Cows with lesions or swellings	( )	[28]
	Cows lying outside lying area	(−)	[27]
	Cows with nasal discharge	(−)	[27]
	Cows with vulvar discharge	(−)	[27]
	Qualitative behaviour assessment	{+}	[27]
	Time needed to lie down	{+}	[27]
FFA (mmol/100 g)	Cows with nasal discharge	(−)	[27]
	Number (length) of drinkers	(−)	[27]
	Time needed to lie down	(−)	[27]

Abbreviations: DIM = days in milk, FFA = free fatty acids. ^1^ Type of associations indicated by ( ) for univariable associations and { } for multivariable associations. ^2^ Direction of associations indicated by + for positive associations, − for negative associations, and ~ for nonlinear associations, empty brackets indicate that no direction was given.

**Table 12 animals-11-03458-t012:** Associations between data-based variables (DBVs) related to cow fertility and animal-based measurements (ABMs) extracted from four scientific publications.

DBV	ABM	Type ^1^ and Direction ^2^	Reference
Fertility Variables Concerning Dairy Cows			
Average calving interval (days)	Lean or very lean cows	( )	[28]
Average calving interval (days)		{+}	[25]
Average expected calving interval (days)		{−}	[27]
Average expected calving interval (days)		( )	[28]
CV calving interval		{−}	[25]
Calving to first service interval (days)		{~}	[27]
Average services per cow		{+}	[27]
Average services per cow		( )	[28]
Cows with >2 services (%)		( )	[28]
Non-return rate 56 days (%)		( )	[28]
Non-return rate 56 days (%)	Lean or very lean cows ≥7.0%	{+}	[28]
Average calving interval (days)		{~}	[28]
Average services per cow	Moderately lame cows	{−}	[27]
Calving to first service interval (days)		{~}	[27]
Average calving interval (days)	Severely lame cows	( )	[28]
Average expected calving interval (days)		( )	[28]
Cows with late ongoing AIs, >120 days (%)		{+}	[25]
Average calving interval (days)	Cows with injuries or inflammation	(−)	[25]
CV calving interval		(−)	[25]
Average services per cow	Cows with lesions or swellings	( )	[28]
Cows with >2 services (%)		( )	[28]
Cows with late ongoing AIs, >120 days (%)	Lean young stock	{+}	[25]
Live born calves per AI series (%)	Lean calves	(−)	[25]
Average expected calving interval (days)	Cows with dirty hindquarter	{~}	[27]
Average services per cow	Cows with dirty legs	{−}	[27]
Cows with late ongoing AIs, >120 days (%)	Dirty young stock	{+}	[25]
Cows with late beginning of AIs, >70 days (%)		(−)	[25]
Calving to first service interval (days)	Cows lying outside lying area	{~}	[27]
Average services per cow	Cows with diarrhoea	{+}	[27]
Non-return rate 56 days (%)	Frequency of displacements	( )	[28]
Fertility Variables Specific for Heifers			
Heifers not bred for >17 months (%)	Lean calves	(+)	[25]
	Lean young stock	{+}	[25]
	Lean or very lean cows	(+)	[25]
	Dirty calves	{+}	[25]
Age at first calving (90 days)	Integument alterations tarsus	(−)	[31]
Average number of AIs per heifer	Lean young stock	(−)	[25]

Abbreviations: AI = artificial insemination, CV = coefficient of variation. ^1^ Type of associations indicated by ( ) for univariable associations and { } for multivariable associations. ^2^ Direction of associations indicated by + for positive associations, − for negative associations, and ~ for nonlinear associations, empty brackets indicate that no direction was given.

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
