# Peer review of "Data-Based Variables Used as Indicators of Dairy Cow Welfare at Farm Level: A Review"

_animals, 2021, doi:10.3390/ani11123458_

Round 1

Reviewer 1 Report

- Do you mean „dairy cows / cattle welfare”?
- I am not sure whether the work being assessed should be treated as a typical review article. For example, Chapters 2 and 3 generally deal with the methodology for analyzing 13 publications that are specifically related to the manuscript. I leave the decision on the qualification of the manuscript to the Editor.
- Lines 95-96: What criteria?
- Lines 171-175: There are no specific proposals for solving the indicated problem in the future in scientific research and / or in the practice of milk production.
- Chapter 5 is very extensive, so it is advisable to summarize the chapter at its end (as in the case of Chapter 4). Also, individual subsections (5.2., 5.3.) Should contain a few-sentence summary / conclusion.
- Subchapter title 5.2.1. should be more specific in relation to the title of chapter 5.2.2.
- Lines 621-625: Not surprisingly, a fundamental lesson for the future is the need to standardize methods for assessing DBV and livestock welfare. It seems that even a cursory knowledge of the state of affairs would lead to such a postulate without the need to read this work. Meanwhile, due to the quite analytical approach of the authors of the reviewed manuscript, it would also be necessary to propose further more particular activities for the future in the aforementioned standardization of the DBVs - animal welfare relationship. Therefore, I expect specific suggestions / guidelines from the authors regarding future research and breeding / production practice.

Author Response

Thank you for your feedback on our manuscript. In the following we would like to respond to your comments and questions:

1.1) Do you mean „dairy cows / cattle welfare”?

Thank you very much for your comment. In this review we have focused on the welfare of dairy cows at farm level. We have adjusted the title accordingly.

1.2) I am not sure whether the work being assessed should be treated as a typical review article. For example, Chapters 2 and 3 generally deal with the methodology for analyzing 13 publications that are specifically related to the manuscript. I leave the decision on the qualification of the manuscript to the Editor.

We refer to our work as review, because the basis for our aim was to summarize available literature with respect to databased indicators to assess dairy cow welfare. As our aim is extremely narrow (see point 1.3), the literature search yielded only 13 publications, which we here review and discuss in the context of a broader set of literature.

1.3) Lines 95-96: What criteria?

Thank you for this remark. To be included in the review, the work had to fulfil two criteria: on the one hand, variables based on routine herd data or records had to be evaluated. On the other hand, the relationship between the data variables and animal welfare at herd or farm level had to be investigated. We have added a description of the criteria in the review.

1.4) Lines 171-175: There are no specific proposals for solving the indicated problem in the future in scientific research and / or in the practice of milk production.

To our knowledge, only two different approaches (three studies) have investigated whether DBVs can be used to detect herds with animal welfare violations. Both approaches showed correlations between DBVs and welfare violations. Due to the small number of studies, research in this area is still in an evaluation phase and should be complemented by further research. Ideally, this research should include broad sets of databased variables to test also DBVs, that have not yet been studied. In order to benefit best from future research, the specific violations of welfare should be described. In addition, studies in different reference populations and thus countries would be desirable. We have added this outlook into possible future research in the review.

1.5) Chapter 5 is very extensive, so it is advisable to summarize the chapter at its end (as in the case of Chapter 4). Also, individual subsections (5.2., 5.3.) Should contain a few-sentence summary / conclusion.

In order to make the described chapters easier to understand and in response to requests from other reviewers, we have shortened Section 5.2.1. In order to avoid prolonging this detailed part, we have omitted a summary at the end of Section 5.2.2, as Section 5.3 is the discussion of the chapter.

1.6) Subchapter title 5.2.1. should be more specific in relation to the title of chapter 5.2.2.

As interest in the use of data variables as indicators of animal welfare continues, we consider it important to provide a brief summary of the studies that have been done so far. In order to allow the reader to situate the results of the reviewed studies (as presented in 5.2.2), we focused on a description of the studies including their methodology in 5.2.1. However, to make chapter 5.2.1 less complex, we have reduced information that is not of fundamental importance for the understanding of the following chapter.

1.7) Lines 621-625: Not surprisingly, a fundamental lesson for the future is the need to standardize methods for assessing DBV and livestock welfare. It seems that even a cursory knowledge of the state of affairs would lead to such a postulate without the need to read this work. Meanwhile, due to the quite analytical approach of the authors of the reviewed manuscript, it would also be necessary to propose further more particular activities for the future in the aforementioned standardization of the DBVs - animal welfare relationship. Therefore, I expect specific suggestions / guidelines from the authors regarding future research and breeding / production practice.

Thank you for the feedback. We have, in order to concretise our suggestions, emphasised them by restructuring chapter 5.3. In addition, we have added the suggestions to the implications for the future. However, we would like to point out, that such suggestions should always be based on the available scientific knowledge. Given that the body of literature contains only little consistent information, due to the heterogeneous nature of the reviewed studies, it is difficult to synthesize a clear, common direction for the future. We therefore have to remain with our main general conclusion that more research is needed, and that said research needs to adhere to common standards, such that a synthesis can indeed be achieved.

Reviewer 2 Report

The article is interesting, but I recommend greater care with the formal espects (number of tables) and a greater number of scientific works used for the analysis.

Author Response

The article is interesting, but I recommend greater care with the formal espects (number of tables) and a greater number of scientific works used for the analysis.

Thank you for your interest in this review. The aim of the review was to bring together the current state of research on the topic of data-based variables as indicators of animal welfare at herd level. The basis for the review was a systematic literature search. Papers that met the following criteria were included in the review. on the one hand, variables based on routine herd data or records had to be evaluated. On the other hand, the relationship between the data variables and animal welfare at herd or farm level had to be investigated. Using this strategic approach, we identified 13 papers.

The formal aspects of the review were checked against the authors' guidelines in 'Animals' before submission.

Reviewer 3 Report

Data-based variables used as indicators of dairy welfare at farm-level: A Review

This manuscript details the assessment of data based variables in the dairy industry for the use of welfare assessment and understanding at the farm-level. Well written and covers an interesting subject. I would consider removing most of the sentences referring to attempting to harmonise a universal definition of welfare as that is unlikely to happen and I feel reference to this weakens the value of the findings. It is well-known in welfare science that a single definition of welfare or a single methodology of welfare assessment does not exist. In line 48 of the introduction the authors themselves acknowledge the lack of harmonization with the definition of welfare.

A few more detailed comments are below.

Watch use of the hyphen and consider rephrasing as comma’s are more suitable in some locations grammatically.

Introduction

Line 67 Reword and replace ‘good’ – perhaps …an abundance of data…

Line 67 Also consider that data availability isn’t as abundant in all countries so this may limit applicability of any findings

Data-Based Variables as Predictors of Farms Violating Animal Welfare

Line 149 Unclear what ‘conspicuous’ variables means, consider rewording

Line 182 Unconvinced that such a detailed summary of the individual papers adds much to the manuscript – consider removing or shortening all sections which are only a review of the individual papers. For example, section 5.2 holds much more value and doesn’t need section 5.1 to be as detailed to set up the discussion for 5.2 and the following sections. Much more interesting to focus on the individual DVMs compared across the studies than to give a summary of each paper.

Data-Based Variables and Animal-Based Measurements

Line 352 Consider defining or rephrasing ‘time needed to lie down’. Is this the actual time spend lying? If not, how do you calculate how much time the cow needed to lie?

Conclusion

Line 608 Remove only

Line 609 Potentially, future work could involve developing the standardized DBVs based on the raw data collected from the 13 studies mentioned or similar.

Author Response

Data-based variables used as indicators of dairy welfare at farm-level: A Review

  • This manuscript details the assessment of data based variables in the dairy industry for the use of welfare assessment and understanding at the farm-level. Well written and covers an interesting subject. I would consider removing most of the sentences referring to attempting to harmonise a universal definition of welfare as that is unlikely to happen and I feel reference to this weakens the value of the findings. It is well-known in welfare science that a single definition of welfare or a single methodology of welfare assessment does not exist. In line 48 of the introduction the authors themselves acknowledge the lack of harmonization with the definition of welfare.

Thank you for your comments on our review. We agree that there is no universal definition of animal welfare. To better highlight this, we have added an additional explanation on different concepts of animal welfare (Line 95-100). As it was not the intention of the review to advocate standardisation of animal welfare or animal welfare assessments or limit them to specific concepts, we have reworded the relevant passages.

A few more detailed comments are below.

  • Watch use of the hyphen and consider rephrasing as comma’s are more suitable in some locations grammatically.

Thank you very much for your advice. This manuscript underwent detailed English editing, including grammar, before submission. We will be happy to make further adjustments should this be desired in the type-editing.

Introduction

  • Line 67 Reword and replace ‘good’ – perhaps …an abundance of data…

Changed as suggested.

  • Line 67 Also consider that data availability isn’t as abundant in all countries so this may limit applicability of any findings

Thank you for pointing this out. We have adjusted the sentence to clarify that the described wealth of routine herd data applies to EU countries.

Data-Based Variables as Predictors of Farms Violating Animal Welfare

  • Line 149 Unclear what ‘conspicuous’ variables means, consider rewording

The authors of the study described used "conspicuous" to describe variables that were conspicuously high or low in the case herds with animal welfare violations compared to herds without animal welfare violations. To clarify this meaning, the sentence was reworded.

  • Line 182 Unconvinced that such a detailed summary of the individual papers adds much to the manuscript – consider removing or shortening all sections which are only a review of the individual papers. For example, section 5.2 holds much more value and doesn’t need section 5.1 to be as detailed to set up the discussion for 5.2 and the following sections. Much more interesting to focus on the individual DVMs compared across the studies than to give a summary of each paper.

Thank you for your comment. We consider it important to provide a brief summary of the studies in order to allow the reader to situate the results of the reviewed studies (as presented in 5.2.2). Nevertheless, we agree that the chapter was very detailed. To make chapter 5.2.1 less complex, we have reduced information that is not of fundamental importance for the understanding of the following chapter.

Data-Based Variables and Animal-Based Measurements

  • Line 352 Consider defining or rephrasing ‘time needed to lie down’. Is this the actual time spend lying? If not, how do you calculate how much time the cow neededto lie?

Thank you for this comment. The ABM ‘time needed to lie down’ represents the time span of a cow’s lying down movement. We have added the definition to the sentence.

Conclusion

  • Line 608 Remove only

Changed as suggested.

  • Line 609 Potentially, future work could involve developing the standardized DBVs based on the raw data collected from the 13 studies mentioned or similar.

Thank you for pointing this out, we have incorporated your suggestion into the conclusion. However, it is difficult to obtain raw data on such studies, as there are usually confidentiality clauses attached to the agreements to participate in the studies.

Reviewer 4 Report

This manuscript aims to review the literature on DBVs as welfare indicators on dairy farms. The topic is timely, and the paper is generally well-written. In my view, however, it does not sufficiently outline, evaluate, or synthesise the existing literature. Beyond superficial descriptions of associations, there is little consideration of why variables might be related. Unreplicated significant findings are glossed over – whilst scepticism is crucial in science, this leads to the conclusion that the current literature has nothing useful to say. I disagree with this assessment of the state of the field.

Abstract

**L30-31: Unclear what’s meant by “linked to the evidence provided”.

**L33-34: Currently, the abstract’s only conclusion is a call for more research and standardisation, which is fairly generic. What are the “limited conclusions” specific to dairy cow welfare?

Introduction

**L37: What are the “problems of livestock farming”? What is the evidence for them?

**L45: Examples of certification schemes and their requirements would make this point less abstract.

**L49-50: I appreciate the authors’ engagement with the nuance of animal welfare science. However, I think they need to discuss what the most prominent conceptions/dimensions of animal welfare are (i.e. subjective wellbeing, physical health, ability to lead a natural life).

**L57: “widely accepted” by whom? Welfare scientists? Farmers? Government?

**L59-61: Please clarify this sentence.

**L61-62: This sentence suggests that ABMs and DBVs are mutually exclusive, which I disagree with. Also, DBVs are another concept that could use examples to illustrate the point (i.e., info given at L65-67).

**L63: “obtain” > “record”.

**L69: Ref?

**L71: This is a fairly superficial explanation of DBVs. What info do they give us? How does that relate to different conceptions of animal welfare?

**L76: “…categorise the PUBLICATIONS according…”

**L76-78: This intro would be strengthened by a clearer statement of purpose, and outline of the manuscript’s structure.

Design and Results of the Literature Search

**L87: “farms with animal welfare violations according to official welfare inspectors” seems like a pretty circular “welfare definition”.

**L96: Whilst this sounds like a fairly comprehensive literature search, the “criteria” aren’t really outlined.

Data-Based Variables as Predictors of Farms Violating Animal Welfare

**L151-156: Unless I’m missing something, these lines contradict each other. Why did the authors “not consider any of the DBVs or sets to be applicable to identify farms with violations of animal welfare” when there were “significant differences” for all five variables?

**L171-175: But this doesn’t reflect the conclusions of either study, which explicitly did link specific DBVs to welfare outcomes. The fact that different methodologies were employed does not nullify these findings.

Data-Based Variables and Animal-Based Measurements

**L178: If five studies each used only one measurement, how were they “predominantly” ABMs with “only a few “RBMs”?

**L206: To make this paper accessible to a wider audience, I’d recommend trying to avoid statistics jargon (here and throughout).

Milk Yield

**L380: Unless discussing mathematical proofs, please avoid “prove”.

Fertility

**L441: Why was this “especially remarkable”? Please try to remain objective.

Data-based variables without clear patterns in the associations with animal-based measurements

**L476-478: What’s meant by “a demonstrated, clear link”?

**L504-509: This is great! The manuscript could use much more such critical examination of the literature and considerations of why findings are inconsistent.

Author Response

This manuscript aims to review the literature on DBVs as welfare indicators on dairy farms. The topic is timely, and the paper is generally well-written. In my view, however, it does not sufficiently outline, evaluate, or synthesise the existing literature. Beyond superficial descriptions of associations, there is little consideration of why variables might be related. Unreplicated significant findings are glossed over – whilst scepticism is crucial in science, this leads to the conclusion that the current literature has nothing useful to say. I disagree with this assessment of the state of the field.

  • Thank you very much for your feedback on the manuscript. Our aim was to extract information on which data based indicators are repeatedly shown as predictors for animal welfare in dairy cattle, irrespective of underlying influencing factors. Put otherwise, we wanted to show for which indicators the value of the information is best confirmed according to the current state of knowledge. We are aware, that each significant result merits its discussion on whether it may be applicable in general or only under certain conditions. However, we believe such discussion to be best placed within the publications of the respective studies. In our opinion the best way to show applicability of the indicators is not by discussing the hypothetical extrapolation from a single significant result in limited experimental conditions, but to demonstrate repeated significance in various studies with different study conditions. Given that all other reviewers suggested a reduction in the detailed description of results, we decided to maintain our focus on the few replicated findings, instead of discussing each significant result (which would inevitably require a detailed description of the respective study conditions).

Abstract

*L30-31: Unclear what’s meant by “linked to the evidence provided”.          

  • Thank you for this comment. In this case, we grouped the associations by the DBVs and analysed them for repetitions. We changed the wording to clarify the meaning of the sentence.

**L33-34: Currently, the abstract’s only conclusion is a call for more research and standardisation, which is fairly generic. What are the “limited conclusions” specific to dairy cow welfare?

  • Thank you very much for this advice. We have adjusted the phrasing and expanded the abstract to include our main conclusions and suggestions for the future.

Introduction

**L37: What are the “problems of livestock farming”? What is the evidence for them?

  • Thank you for the comment, in keeping with the subject of the review we have changed the sentence to refer specifically to animal welfare issues.

**L45: Examples of certification schemes and their requirements would make this point less abstract.

  • Thank you for this feedback. We mention different approaches that aim to ensure animal welfare in order to show the necessity of animal welfare assessment for different stakeholders. We have decided not to mention specific certifications or labels and their requirements at this point in order to keep the introduction short.

**L49-50: I appreciate the authors’ engagement with the nuance of animal welfare science. However, I think they need to discuss what the most prominent conceptions/dimensions of animal welfare are (i.e. subjective wellbeing, physical health, ability to lead a natural life).

  • Thanks for the advice. We assumed that the readers of the section "Animal Welfare" have a basic knowledge of current animal welfare concepts. To better emphasise the multidimensionality, we have outlined Frasers multidimensional approach and the concept of the Five Freedoms.

**L57: “widely accepted” by whom? Welfare scientists? Farmers? Government?

  • Thank you for pointing this out. We have specified the phrase to clarify that ABMs are well recognised in animal welfare research and the World Health Organisation and the European Food Safety Agency support their use.

**L59-61: Please clarify this sentence.

  • Thank you for the comment, we reworded the sentence.

**L61-62: This sentence suggests that ABMs and DBVs are mutually exclusive, which I disagree with. Also, DBVs are another concept that could use examples to illustrate the point (i.e., info given at L65-67).

  • We agree that DBVs and ABMs are not excluding each other and have adjusted this in the text. In addition, the examples mentioned have been added directly to the explanation of DBVs, as suggested.

**L63: “obtain” > “record”.

  • Changed as suggested.

**L69: Ref?

  • We now cite the relevant EU regulation as a reference.

**L71: This is a fairly superficial explanation of DBVs. What info do they give us? How does that relate to different conceptions of animal welfare?

  • Thank you for pointing this out. By restructuring the text in line 113-118 the concept of DBVs has been presented in a more coherent way, with examples of DBVs and their origin.

**L76: “…categorise the PUBLICATIONS according…”

  • Changed as suggested.

**L76-78: This intro would be strengthened by a clearer statement of purpose, and outline of the manuscript’s structure.

  • Thank you for this comment. Currently, the exact structure of the review is given in lines 352-362. If requested by the editor, we will be happy to provide the structure at the end of the introduction instead.

Design and Results of the Literature Search

**L87: “farms with animal welfare violations according to official welfare inspectors” seems like a pretty circular “welfare definition”.

  • Thank you for the comment. Three of the identified works aimed to identify farms with compromised animal welfare in terms of violations against animal welfare legislation using DBVs. We agree that 'violations of animal welfare ' are not a multidimensional definition of animal welfare., but this work is also related to the welfare of dairy cows. Nevertheless, this work also takes into account a relevant area of animal welfare of dairy cows.

**L96: Whilst this sounds like a fairly comprehensive literature search, the “criteria” aren’t really outlined.

  • Thank you for this remark. To be included in the review, the work had to fulfil two criteria: on the one hand, variables based on routine herd data or records had to be evaluated. On the other hand, the relationship between the data variables and animal welfare at herd or farm level had to be investigated. We have added a description of the criteria we applied to the literature.

Data-Based Variables as Predictors of Farms Violating Animal Welfare

**L151-156: Unless I’m missing something, these lines contradict each other. Why did the authors “not consider any of the DBVs or sets to be applicable to identify farms with violations of animal welfare” when there were “significant differences” for all five variables?

  • The paper cited showed a significant difference in the occurrence of five DBVs between case herds with animal welfare violations and herds without animal welfare violations. However, when the DBVs were used as predictors for the presence of animal welfare violations, the sensitivity or specificity was not sufficiently high to actually be used for animal welfare monitoring.

**L171-175: But this doesn’t reflect the conclusions of either study, which explicitly did link specific DBVs to welfare outcomes. The fact that different methodologies were employed does not nullify these findings.

  • We agree and have amended the text to reflect that the three identified publications have each identified DBVs that are related to violations of animal welfare on farm level.

Data-Based Variables and Animal-Based Measurements

**L178: If five studies each used only one measurement, how were they “predominantly” ABMs with “only a few “RBMs”?

  • Thank you for the enquiry. We have reworded the sentence to make it clear that the studies did not each use a single AMB, but that animal welfare was described using a varying number of specific ABMs.

**L206: To make this paper accessible to a wider audience, I’d recommend trying to avoid statistics jargon (here and throughout).

  • Thank you for pointing this out. We have adjusted the relevant line and also reduced the complexity regarding the statistical methods in chapter 5.1.

Milk Yield

**L380: Unless discussing mathematical proofs, please avoid “prove”.

  • Thank you for the input, we have changed the wording.

Fertility

**L441: Why was this “especially remarkable”? Please try to remain objective.

  • Thank you for the input, we have objectified the wording.

Data-based variables without clear patterns in the associations with animal-based measurements

**L476-478: What’s meant by “a demonstrated, clear link”?

  • Thank you for pointing this out. We have changed the wording to clarify that DBVs showed significant associations with ABMs, but these have not yet been repeatedly shown.

**L504-509: This is great! The manuscript could use much more such critical examination of the literature and considerations of why findings are inconsistent.

  • Thank you for your feedback.

Round 2

Reviewer 1 Report

I hereby express my positive opinion on the introduced corrections and explanations of the Authors of the manuscript and express my acceptance of the publication of the work in Animals.

Kind regards

Reviewer

Author Response

Thank you very much for the evaluation of our manuscript,

Yours sincerely, Barbara Lutz

Reviewer 4 Report

*L43: "extend" > "extent"

*L43-47: Grammar

*L89: "data ARE available"

*L396-397: Unclear what is meant by "the time the lying down process needs". The time it takes? How is the beginning and end of the "lying down process" defined?

*L530: "AMBs" > "ABMs"; "therefor" > "therefore"

*L540: "fined" > "defined"?

*L542: An example or two of these detailed/specific definitions would illustrate the point.

*L609-610: "poor emotional state" may need further explanation.

*L707-708: "cattle violating animal welfare" > "welfare violations in cattle"

*L713-716: Confusing

Author Response

Thank you for your comments. Due to the changes made to the manuscript, the line numbers are no longer cosistent. For a better overview, we have therefore partly included the updated passages in our answers. 

L43: "extend" > "extent"

  • Changed as suggested.

*L43-47: Grammar

  • Thank you for your comment, we corrected the relevant passage: “Future research would benefit from a harmonisation of DBVs and the use of valid measurements that reflect the multidimensionality of welfare. Data sources rarely investigated so far may have potential to provide additional DBVs that can contribute to the monitoring of cow welfare at farm level.”

*L89: "data ARE available"

  • Changed as suggested.

*L396-397: Unclear what is meant by "the time the lying down process needs". The time it takes? How is the beginning and end of the "lying down process" defined?

  • Thank you for this comment. We added the definition of "time needed to lie down" (time from bending of the carpal joint to complete lying down on the lying surface)

*L530: "AMBs" > "ABMs"; "therefor" > "therefore"

  • Changed as suggested.

*L540: "fined" > "defined"?

  • Changed as suggested.

*L542: An example or two of these detailed/specific definitions would illustrate the point.

  • Thank you for pointing this out. We included in the manuscript the calculation of the mortality rate used by Krug et al. (2015): 'incidence of on-farm deaths and emergency slaughter reported in death/100 animal-year at risk'. Together with the information on the calculation of animal-years at risk, the publication gives a clear indication of the cows considered for the mortality rate calculation.

*L609-610: "poor emotional state" may need further explanation.

  • Thank you for the comment. We now refer explicitly to the Qualitative Behaviour Assessment of the WQ: "At the criteria level, the occurrence of agonistic behaviour and a poor emotional state, i.e. lower values in the Qualitative Behaviour assessment of the WQ, were associated with lower milk yield."

*L707-708: "cattle violating animal welfare" > "welfare violations in cattle"

  • Changed in “cattle FARMS violating animal welfare”

*L713-716: Confusing

Thank you for pointing this out. We have restructured the relevant text passage.  "In addition, several DBVs were related to scores of welfare assessments such as WQ criteria, CReNBA areas, and overall score. The evaluation of relationships between DBVs and specific ABMs such as lameness or body condition yielded a large number of associations. In this context, DBVs based on mortality were particularly frequently associated with different ABMs. Owing to varying calculations and the consideration of different age or performance groups, a large number of DBV variants were examined. Together with sometimes missing definitions of the variables used, this led to a limited comparability of the studies. This may be a reason why repeated associations of specific DBVs with ABMs were rare."